# The Relationship between Suicide and Oxidative Stress in a Group of Psychiatric Inpatients

**DOI:** 10.3390/jcm9113462

**Published:** 2020-10-28

**Authors:** Tytus Koweszko, Jacek Gierus, Anna Zalewska, Mateusz Maciejczyk, Napoleon Waszkiewicz, Agata Szulc

**Affiliations:** 1Department of Psychiatry, Faculty of Health Sciences, Medical University of Warsaw, 02-091 Warsaw, Poland; agataszulc@poczta.onet.pl; 2Institute of Psychology, Faculty of Psychology, University of Economics and Human Sciences in Warsaw, 01-043 Warsaw, Poland; j.gierus@vizja.pl; 3Experimental Dentistry Laboratory, Medical University of Bialystok, Medical University of Bialystok, 15-089 Białystok, Poland; azalewska426@gmail.com; 4Department of Hygiene, Epidemiology and Ergonomics, Medical University of Bialystok, 15-089 Bialystok, Poland; mat.maciejczyk@gmail.com; 5Department of Psychiatry, Medical University of Bialystok, 15-089 Białystok, Poland; napoleon.waszkiewicz@umb.edu.pl

**Keywords:** psychiatry, suicide, oxidative stress

## Abstract

Diagnosis of suicide risk is a clinical challenge requiring an interdisciplinary therapeutic approach. Except for psychological explanation of the suicidal mechanism, there is evidence that it is associated with brain chemistry disturbances as oxidative stress. The objective of this study was to explore the role of oxidative stress components in suicidality comparing subjects at different stages of suicide. The study included psychiatric inpatients aged 18–64 (*n* = 48) with different psychiatric diagnoses. Blood specimens were collected from subjects and tested for oxidative stress biomarkers: superoxide dismutase (SOD), dityrozine (DT), oxidative stress index (OSI), glutathione peroxidase (GPx), total antioxidant capacity (TAC trolox), ferric reducing ability of plasma (FRAP), total oxidant status (TOS), catalase (CAT), advanced glycoxidation end products (AGE), NADPH oxidase (NOX), and advanced oxidation protein products (AOPP). The Columbia Severity Suicide Scale (C-SSRS) was used for suicidality assessment. Subjects with a history of suicide ideations over the last three months had significantly higher levels of NOX, AOPP, and OSI. There was no significant relationship to any oxidative stress component levels either with a history of suicide behaviors or with suicide attempts over the last three months. The levels of NOX and AOPP were both positively correlated to the intensity of suicidal thoughts. Moreover, there was a positive correlation between a number of suicide attempts during a lifetime with AGE and DT and negative with CAT. Similarly, the subjects with a history of suicide attempts had significantly higher AGE and DT levels and lower CAT values. The study confirmed that oxidative stress plays an important role in the pathophysiology of suicide and specific oxidative stress measures vary in suicidal and non-suicidal psychiatric inpatients.

## 1. Introduction

Suicide phenomenon is a serious worldwide health problem engaging different aspects of human functioning. Every year, approximately 800,000 people die due to suicide [1]. In 2018, in Poland, 45% more people lost their lives due to suicide than because of car accidents [2]. In psychiatric settings, an adequate diagnosis of suicide risk is a well-known problem being an everyday challenge for clinicians. To prevent suicide, it is important to understand that it is not a single act, but a complex process consisting of several stages such as death wish, suicidal ideations, self-harm behaviors, and final act. From a therapeutic perspective, every stage presents an opportunity to intervene and provide therapeutic support [3].

There is a limited number of reliable and valid suicidal risk assessment tools basically limited to semi-structured interviews, protocols, and psychometric scales. However, there is evidence that suicidal impulses are associated with brain chemistry disturbances as oxidative stress [4]. Hence, neurobiological diagnosis might significantly impact the effectiveness of risk assessment.

According to research studies, oxidative stress is associated with several brain-related disorders including Parkinson, Huntington, and Alzheimer’s disease, neurotoxins, trauma, or stroke. Moreover, it is reported that mental health disorders like major depressive or anxiety disorder [5] and autism spectrum disorder [6] are related to oxidative stress disturbances. Schiavone and colleagues suggest that neurobiological modifications in the central nervous system caused by early life traumatic experiences, early maternal separation, parental divorce, physical violence, sexual or psychological abuse, or exposure to war events, also play a significant role in the pathophysiology of oxidative stress [7].

A growing number of studies refer to markers of oxidative stress and inflammation in chronic mental illnesses as schizophrenia [8]. A fortiori both neuroinflammation and increased oxidative stress contribute to neuroprogression observed in the pathogenesis of major depressive disorder (MDD) [9,10], and are depicted as determinants of suicide [4,11]. Data regarding oxidative stress in terms of mental health disorders, however, still remain unclear and ambiguous. At the same time, high risk of suicide in individuals with mental health disorders and trauma reveals a need to explain the pathophysiological mechanisms of suicide to improve diagnostic methods and understand the process of developing suicidal impulses. Research on familiar transmission of suicidal behaviors mainly indicates the role of childhood exposure to parental loss, parental separation, childhood neglect, and child abuse [12]. Environmental exposure is then similar to one indicated by Schiavone [7].

The objective of this study is to explore the role of oxidative stress components in suicidality comparing subjects at different stages of suicide. We hypothesized that measures of oxidative stress components would be different in the group of psychiatric inpatients in terms of their baseline suicidality. Referring to studies on mood disorder factors, it was hypothesized that most of the chosen biomarkers can differentiate suicidality subgroups from the non-suicidal part of the sample. The pilot design of the study led authors to select biomarkers connected to mood disorders presented in the literature of the field. Hypothesized differences are based on the assumption that suicidality, mood disorders, and environmental exposure to trauma/neglect, as presented by Schiavone, present potential shared “nurture” sources in the nature vs. nurture question.

The authors of the study included subjects with different psychopathological disorders in order to identify the transdiagnostic mechanisms of suicide in terms of brain chemistry. The transdiagnostic approach puts the established diagnostic taxonomy aside and focus on specific factors that might be universal for various mental health conditions and explain the pathology no matter of nosological classifications [13].

## 2. Experimental Section

### 2.1. Design

The study included 48 subjects aged 18–64 admitted to the Clinic of Psychiatry of Faculty of Health Sciences, Medical University of Warsaw. The participants of the study had different psychiatric diagnoses: substance-related and addictive disorders (*n* = 13), schizophrenia spectrum and other psychotic disorders (*n* = 9), depressive disorders (*n* = 11), anxiety disorders (*n* = 10), and personality disorders (*n* = 5). The study group included all sequentially admitted patients, regardless of their psychopathology or suicidality. The detailed description of study group is presented in Table 1. The subjects were divided into groups in terms of severity of previous suicidality: suicide ideations, self -harm behaviors, and suicide attempts. Suicide ideations include thinking about, considering, or planning suicide. Self-harm behaviors are any self-destructive actions, however, are not suicide attempts per se. Suicide attempts comprise intentional actions with the intention of taking one’s own life. Suicide ideations group reported suicidal thoughts, and subjects were included into the sample. Subjects who presented any suicidal or harmful behaviors not attempting suicide were included into the ‘self-harm behaviors’ group. Attempted suicide was an ‘suicide attempts’ group inclusion criterion. Exclusion criteria were a history of neurological or organic disorders, or any present major medical illness. Current use of medication was not an exclusion criterion.

The setting of the study assumed that the procedure started in the first 48 h after the hospital admission. At the first stage, the participants were interviewed by the certified clinical psychologist. In the next phase, the blood specimens were collected from subjects and measured for several redox biomarkers: pro-oxidant enzymes, enzymatic and non-enzymatic antioxidants, total antioxidant/oxidant status, oxidative damage products as well as nitric oxide (NO).

The study was approved by the Bioethics Committee of the Medical University of Warsaw (approval number KB/79/2015).

### 2.2. Means of Assessment

Nosological diagnoses were made according to DSM-5 by the Clinic’s staff based on an interview, psychiatric observation, medical documentation, and psychological testing. Subjects were consulted by a psychiatrist and psychologist, while biological markers were collected by the nursing team. The examination was based on a semi-structured interview, providing data regarding the age, marital status, education, place of residence, source of income, and number of suicide attempts in the past. The Columbia Suicide Severity Rating Scale C-SSRS (Risk Assessment Page) was used for suicidality assessment. The researcher completed obligatory C-SSRS training.

According to research studies, C-SSRS is a reliable and valid diagnostic tool in the identification of suicide risk. The scale is evidence-supported and is a part of a U.S. national and international public health initiative involving the assessment of suicidality. The tool is adapted in 103 languages and has been implemented across many settings. Several versions of the C-SSRS have been developed for clinical practice [14,15,16].

All designations were determined in duplicate samples (unless otherwise specified) and standardized to 100 mg of the total protein. The absorbance/fluorescence was measured using an Infinite M200 PRO multimode microplate reader, Tecan.

#### 2.2.1. Pro-Oxidant Enzymes

NADPH oxidase (NOX) activity was determined by the luminescence assay using lucigenin as a luminophore [17]. One unit of NOX activity was defined as the quantity of enzyme required to release 1 nmol of superoxide anion per 1 min.

#### 2.2.2. Enzymatic and Non-Enzymatic Antioxidants

CAT activity was determined in triplicate samples by measuring the decomposition rate of hydrogen peroxide at 240 nm [18]. One unit of CAT activity was defined as the amount of enzyme that decomposes 1 mmol of hydrogen peroxide per 1 min.

Superoxide dismutase (SOD-1) activity was analyzed spectrophotometrically [19] by measuring the inhibition of adrenaline oxidation at 480 nm. It was assumed that 1 unit of SOD-1 activity inhibits the oxidation of adrenaline to adrenochrome by 50%.

Glutathione peroxidase (GPx) activity was determined spectrophotometrically at 340 nm [20]. The method is based on the reduction of organic peroxides in the presence of NADPH. One unit of GPx activity was assumed to catalyze oxidation of 1 μmol of NADPH for 1 min.

#### 2.2.3. Total Antioxidant/Oxidant Status

Total antioxidant capacity (TAC) was analyzed spectrophotometrically at 660 nm based on the reaction with 2,2-azinobis-3-ethylbenzothiazoline-6-sulfonic acid radical cation (ABTS^+^) [21]. TAC levels were calculated from the calibration curve for Trolox (6-hydroxy-2,5,7,8-tetramethylchroman-2-carboxylic acid).

Total oxidant status (TOS) was determined bichromatically at 560/800 nm in triplicate samples. The method is based on the oxidation of Fe^2+^ to Fe^3+^ ions in the presence of the oxidants contained in the samples [22].

Oxidative stress index (OSI) was calculated by dividing TOS by TAC level and expressed in % [23].

Ferric reducing ability of plasma (FRAP) was analyzed spectrophotometrically at 593 nm using 2,4,6-tripyridyl-s-triazine (TPTZ) [24]. FRAP levels were calculated from the calibration curve for iron (II) sulfate (FeSO_4_).

#### 2.2.4. Oxidative Modification Products

Advanced glycoxidation end product (AGE) content was analyzed spectrofluorimetrically [25]. AGE-specific fluorescence was measured at 350 nm/440 nm. For AGE determination, all samples were diluted 1:50 (*v*:*v*) in phosphate buffered saline, pH 7.2 [26].

Advanced oxidation protein products (AOPP) concentration was analyzed spectrophotometrically [25]. The oxidative capacity of the iodine ion was measured at 340 nm. For AOPP determination, all samples were diluted 1:50 (*v*:*v*) in phosphate buffered saline, pH 7.2.

To detect dityrozine (DT) samples were diluted 1:10 (*v*:*v*) in 0.1 M H_2_SO_4_. Fluorescence at 330/415, 365/480, 325/434, and 95/340 nm was analyzed [27]. All results were normalized to fluorescence of 0.1 mg/mL quinine sulfate [28].

#### 2.2.5. Nitric Oxide

Nitric oxide (NO) level was determined spectrofluorimetrically at 543 nm by measuring the NO_3_^−^ and NO_2_^−^ by the Griess reaction [29].

### 2.3. Statistical Analysis

The conducted statistics included the non-parametric Spearman’s rank correlation coefficient and the Mann–Whitney U (*n* = 48) test due to the lack of normal distribution in the examined samples and presence of rank scales. Spearman’s rho was used to analyze the relations between biochemical values, suicidal thoughts intensity, and number of suicide attempts during an individual’s lifetime. The Mann–Whitney U (*n* = 48) test was used to assess significant differences between oxidative stress measures in the groups divided in terms of the presence/absence of suicidal ideations, self-harm behaviors, and suicide attempts over the last three months. The Statsoft STATISTICA 13.1 software was used for statistical analyses.

## 3. Results

The sample comprised 48 patients hospitalized psychiatrically (mean age = 35.7 (+/−11.4) years; females *n* = 23 and males *n* = 25). Subjects with a history of suicide ideations over the last three months (*n* = 31) had significantly higher levels of oxidative stress index (*p* < 0.05), NOX activity (*p* < 0.001), and advanced oxidation protein products (*p* < 0.002). The result are presented in Figure 1. Furthermore, NOX activity (rho = 0.369, *p* < 0.05) and advanced oxidation protein products level (rho = 0.321, *p* < 0.05) were both positively correlated to suicidal thoughts intensity. There was no significant relation of any oxidative stress component levels either with a history of self-harm or with suicide attempts over the last three months.

A number of suicide attempts during an individual’s lifetime significantly correlated with several oxidative stress measures: positively with advanced glycoxidation end products and dityrozine and negatively with catalase (all at *p* < 0.05). Similarly, the subjects with a history of suicide attempts had significantly higher advanced glycoxidation end products (*p* < 0.05) and dityrozine (*p* < 0.05) levels and lower catalase values (*p* < 0.005).

Significant correlations are presented in Figure 2 and Figure 3.

## 4. Discussion

The first finding in our study was the significant relationship of recent suicidal ideations with oxidative stress component levels (oxidative stress index, NADPH oxidase, and advanced oxidation protein products), which does not appear in further stages of suicide (i.e., self-harm behaviors and suicide attempts). We hypothesize that suicidal ideations differ from suicidal actions not only in terms of behavioral and psychological aspects but also in biochemical ones. According to O’Connor’s Integrated Motivational-Volitional Model of Suicide Behavior, a suicide is the result of the interaction of biology, psychology, environment, and culture. The central point of the model assumes that the factors associated with the appearance of suicidal ideations are separate from the tendencies leading to suicidal acts. In O’Connor’s theory suicidal ideations are specific to motivational stage. Usually, it starts with a sense of failure and humiliation. With poor ability to problem solve and tendency to negatively ruminate, the individual starts to have feelings of persecution. Suicidal ideations are a result of psychological suffering, however, they do not necessarily achieve volitional level. This stage moves from thoughts and internal pain to action, bringing illusive feeling of solution finding [30].

Based on the results of our study, we assumed that at the motivational stage of suicide, the individual’s mental and body condition differs from the later ones. The shift to behavioral action is linked with specific physiological changes. Regaining some psychological and biochemical oxidative stress balance might bring relief. Considering depressive patients with reoccurring self-harm behaviors might offer a wider perspective for understanding such phenomena. Several studies dealing with oxidative stress disturbances in depression revealed that major depressive disorder (MDD) is associated with increased levels of NADPH oxidase [31], advanced oxidation protein products (AOPP) [32], and oxidative stress index (OSI) [33]. Interpretations in the literature of the field that link mentioned biomarkers with mood disorders are related mainly to their contribution to alterations in the L-arginine-NO-cGMP pathway and in platelet function, which can lead to the increased thrombotic risk in MDD. The same literature underlines increased protein oxidation with the formation of AOPP, which appears to be hallmark of MDD and bipolar disorder type I. Finally, the research that reports similar results to presented those hereby, confirmed TAC increase while decreasing TOS and OSI in MMD patients after 12 weeks of antidepressant treatment [31,32,33]. The same has been observed in the current study in patients with different psychiatric diagnoses at the stage of suicidal ideations.

The second finding is the increased level of glycation end products (AGEs) and oxidation damage marker dityrozine (DT) as well as the decreased level of CAT in patients with a history of suicide attempts. Moreover, there is a positive correlation between a number of previous suicide attempts and DT and AGEs and negative with CAT. It was reported that there is a relationship between depression and polymorphisms in genes involved in oxidative pathways including CAT [34]. DT and AGEs are known from studies regarding autism spectrum disorder (ASD). According to Anwar and colleagues, ASD is associated with increased AGE and DT compared to healthy controls [35]. Many studies have confirmed that ASD is strongly associated with depression and self-harm behaviors [36,37,38]. In their review, Richa and colleagues stated that over 50% of ASD individuals suffered from depression [39]. Kato and colleagues suggest that “*ASDs should always be a consideration when dealing with suicide attempts in adults at the emergency room”* [40].

According to DSM-V, persistent deficits in social communication and social interaction across multiple contexts and restricted, repetitive patterns of behavior, interests, or activities are characteristic for ASD, contributing to general adaptational problems. ASD symptoms cause substantial impairment in social, occupational, or other important areas of current functioning [41]. At the same time, according to Ringel’s presuicidal syndrome, constriction in individuals in suicidal crisis next to inhibited aggression and suicidal intents causes several temporary disabilities in social functioning, problem solving, emotional expression, and previous values [42]. Similarities between these two groups are debatable, however, the most important one can be chronic distress. The resemblance of these clinical pictures needs more research in order to identify further significant biochemical similarities between autistic disorder and suicidal disturbance. Thus far, the area remains unclear and an ambiguous suggested direction seems to be promising.

Clinical applications of the presented study are limited nowadays. However, further studies on large samples can bring more data on the mechanisms of the interplay between suicidality and biomarkers. One of the most important challenges of psychiatry can be addressed this way: diagnosis based on the subjective reports of patients. Suicidality in one of the central issues of this problem: suicidal patients do not always confirm suicidal ideations to avoid psychiatric or psychological interventions. If any of the biomarkers are confirmed as sensitive and specific enough, this will lead to effective identification of high-risk groups after hospital admission. More objective markers are the goals of modern psychiatry. Some more general physiological issues might be also important in psychiatric conditions, especially referring to the neuroprotective vs. neurotoxic nature of particular physiological processes. The NADPH oxidase NOX2 enzyme is a protein transferring electrons across biological membranes to produce superoxide. Genetic and pharmacological inhibition of NADPH oxidase enzymes can potentially be neuroprotective and has the potential to diminish damaging aspects of pathology following even ischemic and traumatic brain injury as well as in chronic neurodegenerative disorders [17].

The study has several limitations that need to be pointed out. It is a preliminary study designed prior to the performance of a full-scale research project. In the presented study, 48 subjects with different psychiatric conditions were included so the sample was small. Therefore, further research is needed.

Suicide is a complex phenomenon proceeding on several parallel spheres: psychological, behavioral, social, medical, and biochemical. This is why it needs to be indicated that the results of this study do not explain the whole mechanism. Moreover, the impact of medication given to the subjects was not taken into consideration in the research study data analysis. Furthermore, 13 subjects were diagnosed with substance abuse disorders, which can be related to some degree of withdrawal syndrome during data collection. This can lead to drug-induced oxidative stress in the cells.

The study confirmed our initial hypothesis that the oxidative stress plays an important role in the pathophysiology of suicide and specific oxidative stress measures vary in terms of psychiatric inpatients suicidality.

## Figures and Tables

**Figure 1 jcm-09-03462-f001:**
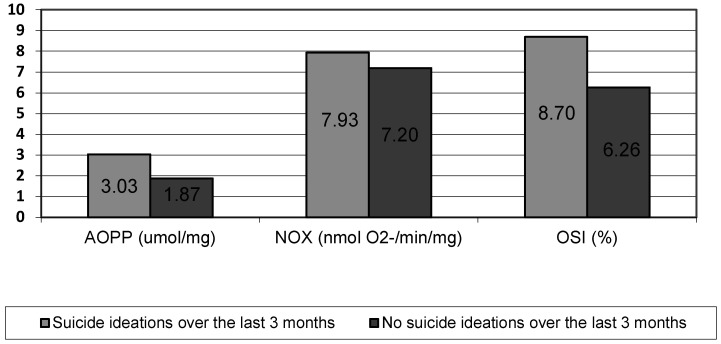
Levels of AOPP, NOX, and OSI in subgroups of suicide ideations (*n* = 31) and no suicide ideations (*n* = 17) over the last three months. AOPP = Advanced Oxidation Protein Products; NOX = NADPH Oxidase; OSI = Oxidative Stress Index.

**Figure 2 jcm-09-03462-f002:**
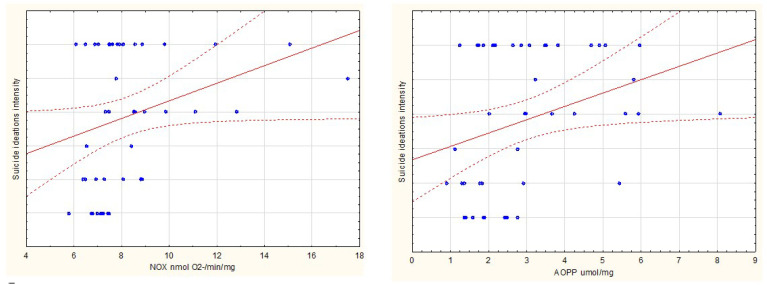
Scatterplots present significant correlations between suicide ideations intensity in the past month and NOX, AOPP (both at *p* < 0.05).

**Figure 3 jcm-09-03462-f003:**
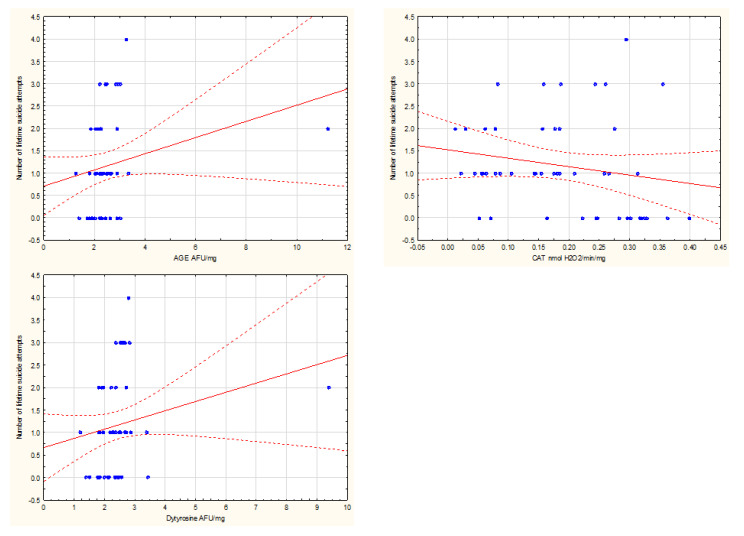
Scatterplots present significant correlations between the number of lifetime suicide attempts and AGE (*p* < 0.05), CAT (*p* < 0.005), and DT (*p* < 0.05).

**Table 1 jcm-09-03462-t001:** Demographics vs. suicidal ideations (SI), self-harm behaviors (SH), and suicidal attempts (SA).

Suicidality	*n*	SI in Last 3 Months	SH in Last 3 Months	SA in Last 3 Months	Number of Lifetime SA
None	Present	None	Present	None	Present	Median	Min	Max
Total	48	17	31	27	21	32	16	1	0	4
**Gender**
Woman	23	10	13	13	10	16	7	1	0	3
Man	25	7	18	14	11	16	9	1	0	4
**Marital status**
Single	24	9	15	10	14	15	9	1	0	3
Married	19	7	12	13	6	13	6	1	0	4
Widowed	1	1	0	1	0	1	0	3	3	3
Divorced	4	0	4	3	1	3	1	1	1	2
**Education**
Primary	7	0	7	2	5	3	4	2	1	4
Vocational	7	4	3	4	3	5	2	1	0	3
Secondary	26	12	14	14	12	16	10	1	0	3
Higher	8	1	7	7	1	8	0	0.5	0	2
**Place of residence**
Village	11	4	7	6	5	8	3	1	0	3
Small town (population <100,000)	20	5	15	12	8	12	8	1	0	4
Big town (population >100,000)	17	8	9	9	8	12	5	1	0	3
**Source of income**
Social benefits	7	4	3	5	2	5	2	1	0	3
Family support	9	4	5	6	3	8	1	1	0	2
Job	32	9	23	16	16	19	13	1	0	4
**Diagnosis**
Substance-related and addictive disorders	13	5	8	7	6	8	5	2	0	4
Schizophrenia spectrum and other psychotic disorders	9	6	3	7	2	9	0	1	0	3
Depressive disorders	11	2	9	7	4	7	4	1	0	3
Anxiety disorders	10	3	7	5	5	5	5	1	0	2
Personality disorders	5	1	4	1	4	3	2	1	1	3
**Lifetime suicide attempt**
None	15	9	6	15	0	15	0			
At least one	33	8	25	12	21	17	16

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
