# Peer review of "The Relationship between Suicide and Oxidative Stress in a Group of Psychiatric Inpatients"

_jcm, 2020, doi:10.3390/jcm9113462_

Round 1

Reviewer 1 Report

The goal of the study was to examine oxidative stress biomarkers in varying degrees of suicidality. Specific oxidative stress biomarkers including NOX, AOPP, and OSI levels, which were higher in subjects with suicide ideations in the past 3 months as compared to those without suicide ideations. Furthermore, NOX and AOPP levels were positively correlated with suicidal thoughts intensity. No differences were observed with suicide behaviors or suicide attempts. However, AGE and DT levels were positively corelated with lifetime number of suicide attempts while CAT levels were negatively correlated. The study findings suggest that specific oxidative stress biomarkers may be associated with occurrence of suicide ideation in the recent pass, and suicide ideation intensity while other oxidative stress biomarkers are related to lifetime suicide attempts. While this study may explicate certain associations between oxidative stress biomarkers and suicidality, my enthusiasm is dampened by the study groups are not well-defined, and the study results section does not match the intended statistical analyses plan (described further under major comments below), suboptimal presentation of results and a poor discussion.  As such, the discussion will also need to be thoroughly revised after re-analyzing the data. 

Major comments:

  1. How were suicidal behaviors defined? Are they distinctly different from suicide attempts in this study?
  2. How many subjects had suicidal ideations versus no suicidal ideations in past 3 months? Similarly, how many in subset of individuals in the suicidal ideation versus no suicidal ideations in past 3 months had lifetime attempts? It is unclear how many subjects were in the groups analyzed with the Mann-Whitney U tests. C-SSRS asks for suicidal ideation in the last month, not 3 months.
  3. How many subjects with suicidal ideation in the past 3 months had a lifetime suicide attempt? If so, would that contribute to the findings? How many control subjects were included in the study without any current or past suicidality (i.e. suicidal ideation, behavior and/or attempts)? Each group should be clearly and explicitly defined.
  4. The authors state psychiatric diagnoses was made by ‘Clinic’s staff’, who do they refer to? Psychiatrist, psychologist, nurse, social worker?
  5. Please discuss strengths and limitations of the study methods and findings.
  6. Please describe the potential implications or clinical applications of the findings.
  7. Were the participants that were diagnosed with substance-related or addictive disorders currently going through withdrawal? How would lifetime or recent substance use be related to the findings or null findings? Smoking, drinking, medical conditions? Medications? All this should, along with demographic characteristics of each group should be presented in a Table, classic Table 1 format.
  8. Please discuss further the reasons why NADPH oxidase, AOPP, and OSI levels were found to be different? Why would it be related to MDD specifically?
  9. The hypothesis could be better derived. Specifically, why were these 11 biomarkers chosen?
  10. Certain medications may also be linked to increase (or decrease) in oxidative stress components. May want to look at subject medication lists and explore this possibility in the discussion.
  11. There is limited discussion of the implications of the authors’ findings. May want to explain more about the physiologic and pathophysiologic significance of oxidative stress in the body
  12. The authors need to revise the statement that autistic patients are similar to suicidal patients, these two types of patients have very different clinical presentations.
  13. Some statement are hard to follow, for example: Line 49: ‘Hence, neurobiological diagnosis might significantly impact on the effectiveness of risk assessment. The manuscript needs a thorough revision by a native English writer for syntax, organization and grammar.

Minor comments:

  1. Consider this alternate title: The relationship between suicide and oxidative stress in a group of psychiatric inpatients.
  2. Please reference supplementary materials in manuscript.
  3. Please recheck acronyms. For example, catalase (CAT) is spelled out twice in lines 188 and 192, and “CAT’ appears in line 190.
  4. In the caption of Figure 1, may want to include the full terms associated with the abbreviations AOPP, NOX and OSI (e.g. AOPP = advanced oxidation protein products). This way the graph can be fully understood even as a standalone figure, separate from the text.
  5. How many subjects had suicidal ideation in the last three months vs. no suicidal ideation in the last three months? Include these N values in Figure 1 caption.
  6. On line 30, “NOx” should be “NOX”.
  7. Why are lines 196-197 italicized?
  8. Since AGEs and DT changes seem to correspond to both the current study sample and ASD, would these be implicated in social functioning impairments and thus increase likelihood for suicide?

Author Response

Response to Reviewer 1 Comments

We are pleased to respond to the Reviewer 1. Thank you for your comments! The article is hopefully improved now in sections and paragraphs advised to be corrected or re-formulated.

Changes as presented below were made in the manuscript:

Major comments:

Point 1: How were suicidal behaviors defined? Are they distinctly different from suicide attempts in this study?

Response 1: “Suicidal behaviors” have been replaced with “self-harm behaviors” which is closer to English terminology. We meant any behaviors, that are self-harmful, however they were not suicide attempt per se (subjects were not motivated with death as a goal). Line 93-99.

Point 2: How many subjects had suicidal ideations versus no suicidal ideations in past 3 months? Similarly, how many in subset of individuals in the suicidal ideation versus no suicidal ideations in past 3 months had lifetime attempts? It is unclear how many subjects were in the groups analyzed with the Mann-Whitney U tests. C-SSRS asks for suicidal ideation in the last month, not 3 months.

Response 2: These questions have been addressed in the Table 1 to give a clearer view of above mentioned data. Places where U test is mentioned were supplemented with additional information (N=…).

C-SSRS Risk Assessment Page gives possibility to collect data about last 3 months:

https://cssrs.columbia.edu/documents/risk-assessment-page/

Point 3: How many subjects with suicidal ideation in the past 3 months had a lifetime suicide attempt? If so, would that contribute to the findings? How many control subjects were included in the study without any current or past suicidality (i.e. suicidal ideation, behavior and/or attempts)? Each group should be clearly and explicitly defined.

Response 3: Table 1 shows now (crosstable) how suicidal ideations relate to lifetime suicidal attempts.

We assumed that there is no possibility to find persons without any lifetime suicidality. It is assumed generally, that every single person has particular situations in his/her life their lives, which is accompanied with some suicidal ideations. We define each group in lines 93-99.

Point 4: The authors state psychiatric diagnoses was made by ‘Clinic’s staff’, who do they refer to? Psychiatrist, psychologist, nurse, social worker?

Response 4: We clarified this question as follows: “Subjects were consulted by psychiatrist and psychologist, while biological markers were collected by the nursing team (lines 114-116).”

Point 5: Please discuss strengths and limitations of the study methods and findings.

Response 5: The limitation paragraph has been added to Discussion section (lines 295-304).

Point 6: Please describe the potential implications or clinical applications of the findings.

Response 6: We rather treat our research as a pilot study, so clinical applications are very limited now. However, we expressed some hopes in lines 300-309.

Point 7: Were the participants that were diagnosed with substance-related or addictive disorders currently going through withdrawal? How would lifetime or recent substance use be related to the findings or null findings? Smoking, drinking, medical conditions? Medications? All this should, along with demographic characteristics of each group should be presented in a Table, classic Table 1 format.

Response 7: 13 of the participants were diagnosed with substance abuse disorders, so this factor can be potential confounder. This can lead to drug-induced oxidative stress in the cells. This has been addressed in the limitation paragraph. We assumed our study, is in fact, a pilot study one. Data about smoking, substance use, and medication are very important, however, in this sample (N=48) probably it would not lead to any valid conclusions.

Point 8: Please discuss further the reasons why NADPH oxidase, AOPP, and OSI levels were found to be different? Why would it be related to MDD specifically?

Response 8: We addressed shortly this issue basing on selected literature.

Point 9: The hypothesis could be better derived. Specifically, why were these 11 biomarkers chosen?

Response 9: We added following sentences to the article: “Referring to studies on mood disorders factors it was hypothesized that most of chosen biomarkers can differentiate suicidality subgroups from non-suicidal part of the sample. The pilot design of the study led authors to select biomarkers that are connected to mood disorders presented in the literature of the field. Hypothesized differences are based on assumption that suicidality, mood disorders and environmental exposure to trauma/neglect, as presented by Schiavone,  present potential shared “nurture” sources in the nature vs. nurture question. Lines 73-78.

Point 10: Certain medications may also be linked to increase (or decrease) in oxidative stress components. May want to look at subject medication lists and explore this possibility in the discussion.

Response 10: Due to pilot nature of the study we didn’t want to take too many variables under consideration. Generally psychiatric medication is linked to reduction of SOD. Antidepressants are usually linked to reduction of some other parameters. Because of modest data in the field it is was hard to make strong assumptions.

Point 11: There is limited discussion of the implications of the authors’ findings. May want to explain more about the physiologic and pathophysiologic significance of oxidative stress in the body

Response 11: This question has been addressed in the discussion section (paragraph “Clinical applications…”). Lines 286-299.

Point 12: The authors need to revise the statement that autistic patients are similar to suicidal patients, these two types of patients have very different clinical presentations.

Response 12: We changes these expressions hopefully to the more balanced ones.

Point 13: Some statement are hard to follow, for example: Line 49: ‘Hence, neurobiological diagnosis might significantly impact effectiveness of risk assessment. The manuscript needs a thorough revision by a native English writer for syntax, organization and grammar.

 Response 13: The language revision was conducted

Minor comments:

Point 1: Consider this alternate title: The relationship between suicide and oxidative stress in a group of psychiatric inpatients.

Response 1: We changed the title including “a”.

Point 2: Please reference supplementary materials in manuscript.

Response 2: We decided to supplement the article with database.

Point 3: Please recheck acronyms. For example, catalase (CAT) is spelled out twice in lines -188 and 192, and “CAT’ appears in line 190.

Response 3: Acronyms are hopefully unified now.

Point 4: In the caption of Figure 1, may want to include the full terms associated with the abbreviations AOPP, NOX and OSI (e.g. AOPP = advanced oxidation protein products). This way the graph can be fully understood even as a standalone figure, separate from the text.

Response 4: We supplemented Figure 1 with full terms of abbreviations.

Point 5: How many subjects had suicidal ideation in the last three months vs. no suicidal ideation in the last three months? Include these N values in Figure 1 caption.

Response 5: We address this comment in the title of the Figure 1.

Point 6: On line 30, “NOx” should be “NOX”.

Response 6: We corrected as suggested by the Reviewer.

Point 7: Why are lines 196-197 italicized?

Response 7: We corrected as suggested by the Reviewer.

Point 8: Since AGEs and DT changes seem to correspond to both the current study sample and ASD, would these be implicated in social functioning impairments and thus increase likelihood for suicide?

Response 8: This might be too strong thesis. We weakened our claims in this field – now we rather suggest that both groups can experience chronic adaptation problems and chronic distress.

Reviewer 2 Report

This is an interesting article, but the number of subjects is low and there is a mixture of different diagnoses. I think the authors should discuss these limitations. A future study could explore if different diagnoses have different results concerning suicidality and the studied biomarkers. It would also be of interest to furhter study the group of subjects with a history of suicide attempts and present suicidal ideation. I would also like the authors to better define the concepts of oxidative stress components, oxidative stress measures and the different substances they have analysed, and to which category they belong, as this is somewhat unclear in the text.

Author Response

Response to Reviewer 2 Comments

We are pleased to respond to the Reviewer 2. Thank you for your comments! The article is hopefully improved now in the sections and paragraphs advised to have been corrected or re-formulated. Changes as presented below were made in the manuscript:

Point 1: This is an interesting article, but the number of subjects is low and there is a mixture of different diagnoses. I think the authors should discuss these limitations. A future study could explore if different diagnoses have different results concerning suicidality and the studied biomarkers.

Response 1: We included the limitations section in the discussion (lines 300-309).

Point 2: It would also be of interest to further study the group of subjects with a history of suicide attempts and present suicidal ideation.

Response 2: The subjects with a history of suicide attempts and present suicidal ideation were presented in the Table 1 (line 103).

Point 3: I would also like the authors to better define the concepts of oxidative stress components, oxidative stress measures and the different substances they have analysed, and to which category they belong, as this is somewhat unclear in the text.

Response 3: The concepts of oxidative stress components were presented in the discussion section. 

Reviewer 3 Report

Review comments

The authors have investigated whether oxidative stress is linked to suicidal thought and /or behaviour in various psychiatric patient groups including major depressive disorder, schizophrenia and anxiety disorders. The research question of the article is interesting, however there are several issues that must be addressed.

General:

-Numerous English language/wording problems appear in the manuscript, which distract from or even distort the conveyed message. I recommend a professional language revision service. For instance, line 20: “in different stages of suicide process”: this suggests that the same people were assessed several times while the process of suicide was happening, which is surely not what the authors aim to express.

Introduction:

Overall, the introduction needs to do a better job in introducing Oxidative Stress specifically for mental disorders, and make clear why the authors focus on suicide risk.  While the authors make clear that suicide has a large societal impact, it is not entirely clear why the authors chose suicide risk specifically for their research.

To improve this, I suggest referring to previous articles discussing similar topics and highlight why not choosing a specific psychopathology is useful (was the reason to identify transdiagnostic mechanisms?).

How could the hypothesized findings link back to inflammation, which has recently been found to be associated to suicide risk? The authors already allude to this, but it needs to be made clearer in my opinion. It would help if specific hypothesis were formulated (i.e. state specifically in which direction they should differ and what exactly is expected to differ in function of which parameter).

Why did the authors choose these specific oxidative stress measures?

Some references that could be named in the introduction and are linked to oxidative stress and suicidal behaviour are below:

E.g.

Batty, G. D., Bell, S., Stamatakis, E., & Kivimäki, M. (2016). Association of systemic inflammation with risk of completed suicide in the general population. JAMA psychiatry73(9), 993-995.

Vargas, H. O., Nunes, S. O. V., de Castro, M. P., Bortolasci, C. C., Barbosa, D. S., Morimoto, H. K., ... & Berk, M. (2013). Oxidative stress and lowered total antioxidant status are associated with a history of suicide attempts. Journal of affective disorders150(3), 923-930.

Lindqvist, D., Dhabhar, F. S., James, S. J., Hough, C. M., Jain, F. A., Bersani, F. S., ... & Rosser, R. (2017). Oxidative stress, inflammation and treatment response in major depression. Psychoneuroendocrinology76, 197-205.

Methods section:

Design

-Line 72 “The subjects were divided into groups in terms of severity of 73 previous suicidality: suicide ideations, suicide behaviors and suicide attempts”: This statement requires some more explanation:

The patients were admitted, but it is not clear whether any suicidal thoughts in the past 3 months were counted as “suicidal ideation” (and likewise for behaviour). Since suicidal behaviour is the main outcome of the study, this needs to be defined more clearly.

-Please describe the diagnostic procedures a little more in depth (if none other than the clinical diagnosis at admission were performed please state so). Were patients who had substance abuse currently using?

-Were statistical results corrected for multiple testing? I would suggest the Benjamini-Hochberg correction.

-Was blood collected sober and at the same time of the day?Please describe the procedure

Results

-The authors should provide an overview table of the demographics, exact suicide attempts/behaviour in each group, with minimum, maximum and median values present.

-Please provide a figure showing the significant correlations.

-I couldn’t find the number of patients in the statistically compared groups. This is very important, and you should give the number of different psychopathologies in each group.

-Do the authors have IL6 and/or other inflammatory markers available that could be correlated to their results?

Discussion

General: The discussion needs restructuring and rewriting. The discussion does in my opinion not discuss the findings sufficiently.

-The authors start with a nice introduction of the O’Connor’s Integrated Motivational-Volitional Model of Suicide Behavior. It is not clear why the authors think that oxidative stress occurs at all in patients with suicidal ideation and how the actual findings of suicidal ideation as opposed to suicidal behaviour link together? Can these findings be integrated to O’connor’s model?

-the paragraph on ASD and suicidal behaviour is not really related to the findings. It’s interesting to state their idea briefly, but now a considerable part of the discussion focusses on ASD and it is not clear why. It would be more pertinent to discuss other disorders that the authors actually assessed and the functional relevance of the altered oxidative stress markers.

-The discussion misses the whole literature on oxidative stress, inflammation and consequences on the brain which should be elaborated on, see  Schiavone, S. T. E. F. A. N. I. A., Neri, M., Mhillaj, E., Morgese, M. G., Cantatore, S., Bove, M., ... & Cuomo, V. (2016). The NADPH oxidase NOX2 as a novel biomarker for suicidality: evidence from human post mortem brain samples. Translational psychiatry6(5), e813-e813.

Author Response

Response to Reviewer 3 Comments

We are pleased to respond to the Reviewer 3. Thank you for your comments! The article is hopefully improved now in sections and paragraphs advised to have been corrected or re-formulated. Changes as presented below were made in the manuscript:

Point 1: General: Numerous English language/wording problems appear in the manuscript, which distract from or even distort the conveyed message. I recommend a professional language revision service. For instance, line 20: “in different stages of suicide process”: this suggests that the same people were assessed several times while the process of suicide was happening, which is surely not what the authors aim to express.

Response 1: The language revision was conducted. Following the reviewer’s suggestion we clarified the objective of the study in lines 19-21 and 70-71:The objective of this study is to explore the role of oxidative stress components in suicidality comparing subjects at different stages of suicide.”

Point 2: Introduction: Overall, the introduction needs to do a better job in introducing Oxidative Stress specifically for mental disorders, and make clear why the authors focus on suicide risk.  While the authors make clear that suicide has a large societal impact, it is not entirely clear why the authors chose suicide risk specifically for their research.

To improve this, I would suggest referring to previous articles discussing similar topics and highlighting why not choosing a specific psychopathology is useful (was the reason to identify transdiagnostic mechanisms?).

How could the hypothesized findings link back to inflammation, which has recently been found to be associated to suicide risk? The authors already allude to this, but it needs to be made clearer in my opinion. It would help if specific hypothesis were formulated (i.e. state specifically in which direction they should differ and what exactly is expected to differ in function of which parameter).

Response 2: We added sentences regarding inflammation and oxidative stress in terms of chronic mental illnesses.

We added additional paragraph at the end of introduction section (lines 79-83):

“The authors of the study included the subjects with different psychopathological disorders in order to identify the transdiagnostic mechanisms of suicide in terms of brain chemistry. The transdiagnostic approach puts the established diagnostic taxonomy aside and focus on specific factors that might be universal for various mental health conditions and explain the pathology no matter of nosological classifications [13]”

We added new position to bibliography: Dalgleish, T., Black, M., Johnston, D., & Bevan, A. (2020). Transdiagnostic approaches to mental health problems: Current status and future directions. Journal of consulting and clinical psychology, 88(3), 179–195. https://doi.org/10.1037/ccp0000482

We clarified our hypotheses (lines 71-78).

Point 3: Why did the authors choose these specific oxidative stress measures?

Response 3: We added following sentences to the article: “Referring to studies on mood disorders factors it was hypothesized that most of chosen biomarkers can differentiate suicidality subgroups from non-suicidal part of the sample. The pilot design of the study led authors to select biomarkers that are connected to mood disorders in the literature of the field.” (line 73-76)

Point 4: Some references that could be named in the introduction and are linked to oxidative stress and suicidal behaviour are below:

E.g.

Batty, G. D., Bell, S., Stamatakis, E., & Kivimäki, M. (2016). Association of systemic inflammation with risk of completed suicide in the general population. JAMA psychiatry73(9), 993-995.

Vargas, H. O., Nunes, S. O. V., de Castro, M. P., Bortolasci, C. C., Barbosa, D. S., Morimoto, H. K., ... & Berk, M. (2013). Oxidative stress and lowered total antioxidant status are associated with a history of suicide attempts. Journal of affective disorders150(3), 923-930.

Lindqvist, D., Dhabhar, F. S., James, S. J., Hough, C. M., Jain, F. A., Bersani, F. S., ... & Rosser, R. (2017). Oxidative stress, inflammation and treatment response in major depression. Psychoneuroendocrinology76, 197-205.

Response 5: We supplemented the bibliography with additional positions:

 Lindqvist, D., Dhabhar, F. S., James, S. J., Hough, C. M., Jain, F. A., Bersani, F. S., ... & Rosser, R. (2017). Oxidative stress, inflammation and treatment response in major depression. Psychoneuroendocrinology76, 197-205.

Batty, G. D., Bell, S., Stamatakis, E., & Kivimäki, M. (2016). Association of systemic inflammation with risk of completed suicide in the general population. JAMA psychiatry73(9), 993-995.

Sawa A, Sedlak TW. Oxidative stress and inflammation in schizophrenia. Schizophr Res. 2016 Sep;176(1):1-2. doi: 10.1016/j.schres.2016.06.014. Epub 2016 Jul 6. PMID: 27395767.

Vargas et al. was already in the bibliography before the revision.

Point 6: Line 72 “The subjects were divided into groups in terms of severity of 73 previous suicidality: suicide ideations, suicide behaviors and suicide attempts”: This statement requires some more explanation:

The patients were admitted, but it is not clear whether any suicidal thoughts in the past 3 months were counted as “suicidal ideation” (and likewise for behaviour). Since suicidal behaviour is the main outcome of the study, this needs to be defined more clearly.

Response 7: “Suicide ideations include thinking about, considering, or planning suicide. Self-harm behaviors are any selfdestuctive actions, however, are not suicide attempts per se. Suicide attempts comprise intentional actions with intention of taking one’s own life. Suicide ideations group reported suicidal thoughts, and subjects were included into the sample. Subjects who presented any suicidal or harmful behaviors not attempting suicide were included into ‘self-harm behaviors’ group. Attempted suicide was an ‘suicide attempts’ group inclusion criterion. (lines 93-99)

Point 8: Please describe the diagnostic procedures a little more in depth (if none other than the clinical diagnosis at admission were performed please state so). Were patients who had substance abuse currently using?

Response 8: The patients were diagnosed at the admission to the hospital. Then the subjects were consulted by psychiatrist and psychologist to verify the initial diagnosis. The biological specimens were collected by nursing team.

13 of the participants were diagnosed with substance abuse disorders, so this factor can be potential confounder. This can lead to drug-induced oxidative stress in the cells. This has been addressed in the limitation paragraph.

Point 9: Were statistical results corrected for multiple testing? I would suggest the Benjamini-Hochberg correction.

Response 9: We used the Bonferroni correction for multiple testing. Some of the results turn out nonsignificant. It confirms that the study needs to be conducted on bigger group of subjects. However results concerning AOPP, NOX, CAT were still significant.

Point 10: Was blood collected sober and at the same time of the day? Please describe the procedure

Response 10: Blood specimens were collected by certified nurse in the morning before breakfast. The subjects were sober.

Point 11: Results: The authors should provide an overview table of the demographics, exact suicide attempts/behaviour in each group, with minimum, maximum and median values present.

Response 11: The data was presented in the Table 1 (line 103).

Point 12: Please provide a figure showing the significant correlations.

Response 12: We presented significant correlations on scatterplots (lines 189-192 and lines 212-230).

Point 13: I couldn’t find the number of patients in the statistically compared groups. This is very important, and you should give the number of different psychopathologies in each group.

Response 13: The number of patients in the compared groups is presented in Table 1 (line 103).

Point 14: Do the authors have IL6 and/or other inflammatory markers available that could be correlated to their results?

Response 14: IL1 and IL6 values were measured however there were not included in this study.

Point 15: The discussion needs restructuring and rewriting. The discussion does in my opinion not discuss the findings sufficiently.

Response 15: The discussion was restructured and rewritten.

Point 16: The authors start with a nice introduction of the O’Connor’s Integrated Motivational-Volitional Model of Suicide Behavior. It is not clear why the authors think that oxidative stress occurs at all in patients with suicidal ideation and how the actual findings of suicidal ideation as opposed to suicidal behaviour link together? Can these findings be integrated to O’connor’s model?

Response 16: The O’Connors model was presented in the discussion to provide psychological background and supplement the biochemical perspective on suicide. According to Slavish and Auerbach (2018) stress of contemplating suicide increases levels of inflammation. Chronic inflammation as a result of sustained perceptions of social–environmental threat / intense psychological distress can cause oxidative stress. People suffering from suicide ideations – being in motivational stage of O’Connor’s model – suffer from intense psychological pain so the probability of inflammation increases. We assumed that the switch to volitional stage of O’Connor’s model decreases the psychological distress resulting in further changes in terms of oxidative stress components levels. From clinical perspective the depressed patients with intense suicide ideations suffer from strong psychological distress. However, the decision of suicide often brings them relief. Therefore our findings suggest that the first stage of suicide – with suicide ideations (O’Connor motivational stage) – might play a specific role in terms of oxidative stress. We are aware that this conclusion is uncertain, however, it is only an initial one due to preliminary nature of our study. We assume that the research on larger group might allow us to verify our hypothesis.

Reference: Slavich, G. M., & Auerbach, R. P. (2018). Stress and its sequelae: Depression, suicide, inflammation, and physical illness. In J. N. Butcher & J. M. Hooley (Eds.), APA handbook of psychopathology: Vol. 1. Psychopathology: Understanding, assessing, and treating adult mental disorders (pp. 375-402). Washington, DC: American Psychological Association. doi: 10.1037/0000064-016

Point 17: the paragraph on ASD and suicidal behaviour is not really related to the findings. It’s interesting to state their idea briefly, but now a considerable part of the discussion focusses on ASD and it is not clear why. It would be more pertinent to discuss other disorders that the authors actually assessed and the functional relevance of the altered oxidative stress markers.

Response 17: We admit that this might be too strong thesis. We weakened our claims in this field – now we rather suggest both groups can experience chronic adaptation problems and chronic distress. 274-284

Point 18: The discussion misses the whole literature on oxidative stress, inflammation and consequences on the brain which should be elaborated on, see  Schiavone, S. T. E. F. A. N. I. A., Neri, M., Mhillaj, E., Morgese, M. G., Cantatore, S., Bove, M., ... & Cuomo, V. (2016). The NADPH oxidase NOX2 as a novel biomarker for suicidality: evidence from human post mortem brain samples. Translational psychiatry6(5), e813-e813.

Response 18: The discussion and bibliography have been improved.